# Gender and Public Perception of Disasters: A Multiple Hazards Exploratory Study of EU Citizens

**Arturo Cuesta** [1,*]**, Daniel Alvear** [1]**, Antonio Carnevale** [2] **and Francine Amon** [3]

1. Department of Transports Projects and Processes, University of Cantabria, Ave. Los Castros, s/n, 39005 Santander, Spain
2. CyberEthics Lab., Via Giuseppe Vasi 18/A, 00187 Rome, Italy
3. RISE Research Institutes of Sweden, P.O. Box 857, 50115 Borås, Sweden
* Correspondence: arturo.cuesta@unican.es

**Abstract: Aim:** To explore gender influence on individual risk perception of multiple hazards and personal attitudes towards disaster preparedness across EU citizens. **Method:** An online survey was distributed to 2485 participants from Spain, France, Poland, Sweden and Italy. The survey was divided into two parts. The first part examined perceived likelihood (L), perceived personal impact (I) and perceived self-efficacy (E) towards disasters due to extreme weather conditions (flood, landslide and storm), fire, earthquake, hazardous materials accidents, and terrorist attacks. The overall risk rating for each specific hazard was measured through the following equation $R = (L \times I)/E$ and the resulting scores were brought into the range between 0 and 1. The second part explored people's reactions to the Pros and Cons of preparedness to compute the overall attitudes of respondents towards preparation (expressed as a ratio between −1 and 1). **Results:** Although we found gender variations on concerns expressed as the likelihood of the occurrence, personal consequences and self-efficacy, the overall risks were judged significantly higher by females in all hazards ($p < 0.01$). We also found that, in general, most respondents (both males and females) were in favour of preparedness. More importantly, despite the gender differences in risk perception, there were no significant differences in the attitudes towards preparedness. We found weak correlations between risks perceived and attitudes towards preparedness (rho < 0.20). The intersectional analysis showed that young and adult females perceived higher risks than their gender counterparts at the same age. There were also gender differences in preparedness, i.e., females in higher age ranges are more motivated for preparedness than men in lower age ranges. We also found that risk perception for all hazards in females was significantly higher than in males at the same education level. We found no significant differences between sub-groups in the pros and cons of getting ready for disasters. However, females at a higher level of education have more positive attitudes towards preparedness. **Conclusions:** This study suggests that gender along with other intersecting factors (e.g., age and education) still shape differences in risk perception and attitudes towards disasters across the EU population. Overall, the presented results policy actions focus on promoting specific DRR policies and practices (bottom-up participatory and learning processes) through interventions oriented to specific target groups from a gender perspective.

**Keywords:** gender; public perception; multiple hazards; risk perception; preparedness





## 1. Introduction

Between 2000 and 2021 in total 14.189 disasters have occurred worldwide causing around 1.5 million casualties. Of these, 1.633 disasters have occurred in Europe with 169.402 reported casualties [1]. The role people play before, during and after a disaster is of crucial importance. In fact, the active participation of individuals and communities is a principle of the Sendai Framework for Disaster Risk Reduction 2015-2030 (SFDRR) [2]. Bottom-up participatory and learning processes in which citizens can act by themselves

and/or together with emergency services are the suggested mechanisms to improve Disaster Resilience and Response (DRR) [3,4]. The Sendai Framework also recognizes the importance of integrating a gender perspective into all DRR policies and practices. Hence, for effective bottom-up implementation of DRR policies and practices, we need first to understand the differences/similarities of the risks perceived by both women and men and their subsequent attitudes toward preparedness (as a behavioural precursor).

Although disasters affect whole communities, they are not gender neutral as they impact women and men differently. Gender issues (economic, social, and political inequalities) can create specific vulnerabilities for women in disasters [5,6]. Moreover, gender structures shape the roles, experiences, and responsibilities of individuals in disasters [6,7]. The typical gender roles in disasters are described by Enarson [8] and Fothergill [9].

Gender can also be related to risk judgments and attitudes towards safety [10–13]. In this sense, risk perception and preparedness have been the central investigated issues. Some studies directly address gender influence on these subjects and others include gender among other predictors/variables by simply reporting gender "differences". Regardless of the method used, the literature indicates that women in general perceive hazards as being more serious and riskier than men [8,14–17] and that men express more confidence to face disasters [13,18,19]. Researchers have also focused on preparedness by exploring gender among other factors (e.g., race/ethnicity, age, education, etc.). Some studies showed that gender acts as a predictor of preparedness with women being less likely to be prepared than men for specific hazards [20–23] but other studies were not conclusive (e.g., [24,25]).

Risk perception is usually conceptualised as a logical predictor of preparedness. However, the link between these constructs is still not clear [26,27]. Whereas some studies found that risk perception is associated with or predicts preparedness [23,28,29], others did not [30–32]. Furthermore, most previous studies concentrated on unique disasters (past and/or potential) in specific geographical regions with distinct degrees of gender relations/inequalities, or they were conducted for specific communities or groups of people living in the affected areas. Hence most research findings, although useful for regional and local authorities, are context dependent and difficult to generalize to other hazards and areas. For instance, for an EU policy implementation. Another aspect not fully addressed in the literature is the analysis of gender along with intersectional factors (intersectional approach) (Gendered Innovations: http://genderedinnovations.stanford.edu/terms/intersectionality.html, accessed on 10 May 2022) in the context of disaster response as gender identities, norms, relations and attitudes both shape and are shaped by other social attributes.

The aim of this study is to investigate gender influence on risk perception for multiple hazards and attitudes toward preparedness from a regional perspective, i.e., across the EU population. The first question to investigate is, since women are likely to perceive higher risks than men, is it reasonable to think that they are also more motivated for preparedness? Otherwise (i.e., if there is no positive association between risk and preparedness in gender groups) is it reasonable to infer that gender, among other intersectional factors, contributes to shaping people's attitudes towards disasters? Therefore, the present study aims at contributing to current knowledge by analysing datasets from a multinational survey. The collected responses provided the opportunity to explore gender differences/similarities of EU citizens (from Spain, Poland, Sweden, France and Italy).

The main objectives of the current study are listed below:

- To find gender differences/similarities in risk perception and attitudes toward disaster preparedness;
- To investigate whether risk perception is associated with the intent to prepare for disasters in different genders;
- To explore subgroup differences among males and females according to age and educational background.

Datasets produced here not only have scientific value but also have the potential to inform policymakers and first responders for developing risk management policies and

training and communication campaigns, thus improving disaster response and resilience of society as viewed using a gender perspective in Europe.

## 2. Method

The Checklist for Reporting Results of Internet E-Surveys (CHERRIES) was used as a reference to provide exhaustive information on the survey and to facilitate reproducibility [33].

**Design**.—The survey was designed to cover people's risk perceptions and attitudes towards preparedness for disasters. The questions used to investigate these factors are listed in Table 1. To analyze risk perception we focused on three main factors inspired by the Protection Motivation Theory developed by Rogers [34,35] and also applied to disaster research [36,37]. These three factors are: (1) the likelihood of disasters to occur (L), (2) the personally relevant impact if disasters occur nearby (I) and (3) the perceived self-efficacy to face the disasters (E). Each question was asked in relation to extreme weather conditions, fires, earthquakes, hazardous material accidents and terrorist attacks. The rationale for the selection of these hazards was their global relevance in Europe (Table 1): meteorological (storms, extreme temperatures, floods), climatological (wildfires), geophysical (earthquakes), technological (industrial accidents) [1] and terrorism (terror attacks) [38,39].

**Table 1.** Human consequences of selected disasters for the last 20 years in Europe. Sources: EM-DAT [1] and GTD [40].

| Hazard | People Affected | Injuries | Casualties |
|---|---|---|---|
| Extreme weather conditions | 11,540,045 | 35,918 | 154,864 |
| Extreme temperature | 688,787 | 23,350 | 151,884 |
| Flood | 6,852,496 | 8847 | 2134 |
| Storm | 3,998,762 | 3721 | 846 |
| Wildfire | 1,287,245 | 3981 | 538 |
| Earthquake | 594,175 | 4402 | 782 |
| Industrial accident | 18,564 | 4264 | 1323 |
| Terrorist attack * | | 4547 | 642 |

* Bomb and shooting attacks in Western Europe.

In addition, 9-item questions were included to explore the attitudes of males and females towards disaster preparedness: 4 statements for the Pros and 5 statements for the Cons. For simplicity, the statements are expressed as Resilience, Information, Confidence, Assistance for the Pros and Uselessness, Buck-passing, Avoidance, Denial and Cost for the Cons (Table 2).

**Table 2.** Survey questions and the related available answers. * Extreme weather conditions, Fire, Earthquake, Hazardous materials accidents, and Terrorist attacks. ˆ words in parentheses were not included in the questionnaire but are included here to remind the reader of the survey design.

| | Variable | Question | Available Answers |
|---|---|---|---|
| Risk Perception | Likelihood | *How likely do you consider that * will occur nearby?* | On a scale from 1 "Highly unlikely" to 4 "Highly likely" |
| | Impact | *If * occur in your vicinity, what in your view is the impact for you and your family?* | On a scale from 1 "Very low" to 4 "Very high". |
| | Self-efficacy | *Which statement best represents your ability to deal with *.* | On a scale from 1 "I don't know what to do" to 3 "I know what to do" |

**Table 2.** *Cont.*

| | Variable | Question | Available Answers |
|---|---|---|---|
| Attitudes towards preparedness | *Pros* | *Getting ready is worthwhile because:*<br>• *It is easier to get back to normal (Resilience)* ^<br>• *I can have information about what to do (Information)*<br>• *Acting makes me worry less (Confidence)*<br>• *If I am ready, I can help others (Assistance)* | |
| | *Cons* | *Getting ready is not worthwhile because:*<br>• *It won't make a difference (Uselessness)*<br>• *It is not my responsibility*<br>• *(Buck-passing)*<br>• *I would rather not think about bad things happening (Avoidance)*<br>• *It doesn't matter; disasters don't happen where I live (Denial)*<br>• *It takes too much time and effort (Cost)* | On a scale from 1 "Strongly disagree" to 5 "Strongly agree" |

**Ethics**.—The questionnaire was anonymous, and the privacy policy of the individual's posted information was noted (e.g., the purpose of the study, length of time to complete the survey, personal data and data protection, withdrawal rights, etc.). Due to the nature of this study written informed consent was not required. However, respondents were informed about the purpose of the study, and their rights and gave consent to participate by filling in the agreement part of the survey form. This study was approved by the Ethical Committee of the University of Cantabria.

**Development**.—A pilot was conducted involving 56 participants, allowing us the possibility to know whether a designed questionnaire fulfilled the purpose of the study (i.e., the respondents were asked whether the questions were clear and if they interpreted them as expected). The English version of the questionnaire was reviewed by two external experts and then translated into the target languages by native speakers. During the translation process, we paid special attention to achieving semantic, idiomatic, experiential, and conceptual equivalence to the original version. The initial translation into each target language was made by two independent translators per language to detect and resolve subtle differences/discrepancies. Also, the resulting versions were back-translated to ensure the accuracy of the translation. Then, the online prefinal versions were sent again to the translators for checking and final approval. Check-box answers were provided in the questionnaire to reduce the time to answer each item. Different scales were used. We considered a standard 5-point Likert scale (with a neutral option) for the Pros and Cons of preparedness as we wanted to collect enough granularity in opinions and attitudes. For self-efficacy, we reduced the response options using a 3-point Likert scale forcing the respondents to provide two pieces of information (two polar points along with a neutral option) based on the assumption that collapsing data from a longer scale into three-point scales does not diminish the reliability or validity of the resulting scores while enabling to collect clear responses about perceptions of self-efficacy to face disasters. For likelihood and impact, we used a 4-point Likert with no neutral option thus participants were required to form a judgment while reporting the intensity of the direction. Place of residence (village/town/city), education (no studies/primary/secondary/university), age, occupation (self-employed/employee/unemployed/retired/student) and gender (male/female/binary/other) were gathered at the starting section of the questionnaire.

**Survey administration**.—The usability and functionality of the electronic questionnaires were tested before fielding the final versions. A hired survey company sent an email invitation to individuals 2.485 living in the targeted countries. In total, we received 1.047 responses (response rate of 41.13%). Respondents belonged to validated databases and were given a monetary incentive for their participation. The company ensured a level of quality control, before and during the data collection.

The questionnaire had in total of 26 items in addition to the sociodemographic information on the first screen. Items were randomized to prevent biases in responses. Overall, the questionnaire took approximately 10–15 min to complete. The responses (only one per participant) were automatically captured and checked through the online survey system. The timeframe for the data collection was from 1 to 14 November 2020.

**Participants.**—Out of the 1.047 responses 1.2% identified as "non-binary" or "other" rather than "man" or "woman". This "non-binary" group comprised a very small sample size for statistical testing. Therefore, the population sample for the study involved 1.014 respondents (510 who identified as men and 504 who identified as women) from five countries representative of northern (Sweden), southern (Italy and Spain), eastern (Poland) and western (France) regions of Europe. Table 3 displays the characteristics of the surveyed participants. We compared our sample and the sociodemographic characteristics of those surveyed with the Eurostat census data [41]. The Eurostat for adults (aged 20 years and over) shows that 52% of females gave a 2.27% point (pp) difference between our data and the EU population. The age of respondents (20–69 years) was quite representative with an average difference of 4.69% (pp). Yet, there was an over-representation from respondents <29 years (absolute difference of 9.93%) and an under-representation from respondents >60 years (absolute difference of 7.68%). The dwelling type of our sample had absolute differences of 8.8% for cities, 0.3% for towns and 9.2% for rural areas when compared with Eurostat data. Education level (Secondary and University: sample = 91.4% vs. EU population = 79.50%) and occupation (people in the labour force; sample = 69% vs. EU population = 77.10%) had differences but reasonably represented in our study.

**Table 3.** Baseline characteristics of study participants. Significant *p*-values in bold.

| Variable | Overall (*n* = 1.014) | Male (*n* = 510, 50.3%) | Female (*n* = 504, 49.7%) | *p*-Value |
|---|---|---|---|---|
| Age, years (Mean ± SD) | 41 ± 22.7 | 45 ± 15.7 | 37 ± 13.3 | **<0.001** |
| Dwelling type [*n* (%)] | | | | 0.23 |
| City | 480 (47.34) | 248 (24.46) | 232 (22.88) | |
| Town | 348 (34.32) | 179 (17.65) | 169 (16.67) | |
| Rural areas | 186 (18.34) | 83 (8.19) | 103 (10.16) | |
| Country [*n* (%)] | | | | 0.99 |
| France | 207 (20.41) | 107 (10.55) | 100 (9.86) | |
| Italy | 202 (19.92) | 100 (9.86) | 102 (10.06) | |
| Poland | 201 (19.82) | 100 (9.86) | 101 (9.96) | |
| Spain | 203 (20.02) | 103 (10.16) | 100 (9.86) | |
| Sweden | 201 (19.82) | 100 (9.86) | 101 (9.96) | |
| Education level [*n* (%)] | | | | 0.23 |
| No studies | 11 (1.08) | 7 (0.69) | 4 (0.34) | |
| Primary | 76 (7.5) | 41 (4.04) | 35 (3.45) | |
| Secondary | 437 (43.10) | 231 (22.78) | 206 (20.32) | |
| University | 490 (48.32) | 231 (22.78) | 259 (25.54) | |
| Occupation [*n* (%)] | | | | **<0.001** |
| Self-employed | 95 (9.37) | 56 (5.52) | 39 (3.85) | |
| Employee | 535 (52.76) | 270 (26.63) | 265 (26.13) | |
| Unemployed | 146 (14.40) | 43 (4.24) | 103 (10.16) | |
| Retired | 109 (10.75) | 77 (7.59) | 32 (3.16) | |
| Student | 65 (12.72) | 64 (6.31) | 65 (6.41) | |

**Analysis.**—Descriptive statistics are presented as absolute counts and/or percentages for ordinal variables while interval variables are expressed by means (with SD). To measure an individual's risk rating (R) for each of the five hazards we computed the likelihood (L), the personal impact (I) and the perceived self-efficacy (E) through the following equation R = (L × I)/E based on [16]. We assumed that the perceived self-efficacy affects the risk perceived rather than simply considering the perceived likelihood and impact to

measure risk ratings [17]. Hence, the perceived risk is minimized/reduced (or not) by the perceived self-efficacy here assumed as a value between 1 and 3 where 1 is "I don't know what to do", 2 is "I might know what to do" and 3 is "I know what to do". In the first case, self-efficacy does not change the perceived likelihood and impact. In the second case likelihood and impact are reduced by half. In the third case likelihood and impact are reduced by three times. The resulting scores were brought into the range between 0 and 1 for better understanding and further comparison with other datasets. To measure the attitudes toward preparedness, the responses to each item were summed to create composite scores (of Pros and Cons) for each respondent. The resulting scores were also normalized, and the overall attitudes were expressed as a ratio between −1 and 1 that resulted from subtracting the Pros score from the Cons score. Non-parametric methods were used to assess differences between groups: cross-tabulation and Pearson's chi-square for relative frequencies, Wilcoxon rank sum test and Kruskal-Wallis (Dunn's test) for ordinal and interval scales. The JASP statistical program v0.15 was used for statistical tests throughout the entire study (JASP Team, 2021). For all analyses performed in our study, *p*-values < 0.05 were considered statistically significant.

## 3. Results

**Risk perception**.—The variables related to likelihood (L), impact (I) and self-efficacy (E) for multiple hazards are listed in Table 4. There were gender differences when anticipating the occurrence of extreme weather (W =137,559, *p* = 0.03) and fire (W = 138,582, *p* = 0.01) considered less likely by males than females. We also found that gender is associated with the perceived impact of extreme weather (W = 139,124, *p* = 0.01), fire (W = 137,607, *p* = 0.03) and earthquake (W = 141,289, *p* < 0.01) if it occurs nearby. Nevertheless, the item score distributions of the perceived impacts for hazardous materials accidents (W=133,452, *p* = 0.27) and terrorist attacks (W = 131,533, *p* = 0.50) did not differ significantly between males and females. Our results also suggest that males expressed higher perception of their coping abilities than females to face potential hazards: extreme weather conditions ($\chi^2$ = 20.4, *p* < 0.01), fire ($\chi^2$ = 22.45, *p* < 0.01), earthquake ($\chi^2$ = 12.18, *p* < 0.01), hazardous materials accident ($\chi^2$ = 36.60, *p* < 0.01) and terrorist attack ($\chi^2$ = 47.93, *p* < 0.01). Importantly, gender differences were found to be statistically significant (*p* < 0.01) in the overall risk perception with higher scores in females than in males (Table 5).

**Table 4.** Absolute counts of respondents in the perceived likelihood (from 1 = highly unlikely to 4 = highly likely), impact (from 1 = very low to 4 = very high) and self-efficacy (1 = I do not know what to do; 2 = I fairly know what to do; 3 = I know what to do) for extreme weather conditions, fire, earthquake, hazardous material accidents and terrorist attack. *p*-values of the two-sided Wilcoxon rank sum test for likelihood and impact and Chi-Square test for self-efficacy. The significant *p*-value is in bold.

| | Likelihood (L) | | | | Impact (I) | | | | S-Efficacy (E) | | |
|---|---|---|---|---|---|---|---|---|---|---|---|
| | 1 | 2 | 3 | 4 | 1 | 2 | 3 | 4 | 1 | 2 | 3 |
| **Extreme weather** | | | | | | | | | | | |
| Female (*n*) | 49 | 131 | 241 | 83 | 51 | 246 | 163 | 44 | 174 | 286 | 44 |
| Male (*n*) | 67 | 144 | 228 | 71 | 91 | 231 | 147 | 41 | 113 | 332 | 65 |
| *p*-value | | **0.03** | | | | **0.01** | | | | **<0.001** | |
| **Fire** | | | | | | | | | | | |
| Female (*n*) | 31 | 105 | 270 | 98 | 50 | 231 | 165 | 58 | 111 | 309 | 84 |
| Male (*n*) | 46 | 130 | 245 | 89 | 73 | 228 | 174 | 35 | 71 | 300 | 139 |
| *p*-value | | **0.01** | | | | **<0.01** | | | | **<0.001** | |

**Table 4.** *Cont.*

| | Likelihood (L) | | | | Impact (I) | | | | S-Efficacy (E) | | |
|---|---|---|---|---|---|---|---|---|---|---|---|
| | 1 | 2 | 3 | 4 | 1 | 2 | 3 | 4 | 1 | 2 | 3 |
| **Earthquake** | | | | | | | | | | | |
| Female (*n*) | 170 | 181 | 123 | 30 | 121 | 193 | 136 | 54 | 218 | 244 | 42 |
| Male (*n*) | 187 | 173 | 114 | 36 | 168 | 176 | 125 | 41 | 175 | 266 | 69 |
| *p*-value | | 0.50 | | | | **<0.01** | | | | **<0.01** | |
| **Hazardous material accident** | | | | | | | | | | | |
| Female (*n*) | 146 | 208 | 128 | 22 | 106 | 185 | 140 | 73 | 366 | 119 | 19 |
| Male (*n*) | 148 | 205 | 129 | 28 | 140 | 151 | 152 | 67 | 278 | 192 | 40 |
| *p*-value | | 0.77 | | | | 0.27 | | | | **<0.001** | |
| **Terrorist attack** | | | | | | | | | | | |
| Female (*n*) | 126 | 195 | 143 | 40 | 117 | 189 | 131 | 67 | 331 | 155 | 18 |
| Male (*n*) | 138 | 175 | 134 | 63 | 136 | 177 | 127 | 70 | 236 | 214 | 60 |
| *p*-value | | 0.59 | | | | 0.50 | | | | **<0.001** | |

**Table 5.** Differences in overall risk perception according to gender. Normalized Mean scores, SD standard deviation [0, 1]. *p*-values of the two-sided Wilcoxon rank sum test. The significant *p*-value is in bold.

| Hazards/Disasters | Male Mean ± SD | Female Mean ± SD | W | *p*-Value |
|---|---|---|---|---|
| Extreme weather | 0.21 ± 0.17 | 0.26 ± 0.19 | 150,839 | **<0.001** |
| Fire | 0.20 ± 0.15 | 0.24 ± 0.16 | 152,388 | **<0.001** |
| Earthquake | 0.15 ± 0.15 | 0.19 ± 0.16 | 144,860 | **<0.001** |
| Hazardous Materials Accident | 0.22 ± 0.19 | 0.25 ± 0.19 | 143,337 | **<0.01** |
| Terrorist attack | 0.23 ± 0.22 | 0.26 ± 0.22 | 143,289 | **<0.01** |

　　　**Attitudes towards preparedness**.—Most respondents were in favour of getting prepared for disasters (Table 6). There were no statistically significant gender differences for Resilience "it is easier to get back to normal", Information "people have information about what to do" and Confidence "taking action makes me worry less" as Pros of preparedness. Interestingly, the importance of preparedness for helping others (i.e., Assistance) was significantly higher for females than males (W = 138,204, *p* = 0.02). Around one-fourth of respondents did not form an opinion on the Cons of preparedness and chose the neutral option "undecided" for Avoidance (28% females; 25% males), Denial (23% females; 25% males) and Cost (22% females; 24% males). No significant gender differences were found for Uselessness "getting ready won't make a difference", Buck-passing "It is not my responsibility", and Cost "It takes too much time, effort, or money". Yet, differences were statistically significant for Avoidance "I would rather not think about bad things happening" (W = 138,848.5, *p* = 0.02) and Denial "It doesn't matter; disasters don't happen where I live" (W = 119,186, *p* = 0.03). However, one of the interesting results that emerged from the data was that gender differences in the composite scores for Pros and Cons of getting ready and the overall attitudes toward preparedness were not statistically significant (Table 7).

**Table 6.** Respondents' reactions to the Pros and Cons of disaster preparedness (from 1 = strongly disagree to 5 = strongly agree). *p*-values of the two-sided Wilcoxon rank sum test. The significant *p*-value is in bold.

| Pros | Score | | | | | Cons | Score | | | | |
|------|-------|---|---|---|---|------|-------|---|---|---|---|
| | **1** | **2** | **3** | **4** | **5** | | **1** | **2** | **3** | **4** | **5** |
| Resilience | | | | | | Uselessness | | | | | |
| Female (%) | 3 | 5 | 20 | 48 | 25 | Female (%) | 43 | 32 | 13 | 10 | 2 |
| Male (%) | 1 | 6 | 22 | 46 | 26 | Male (%) | 38 | 32 | 17 | 11 | 2 |
| *p*-value | | | 0.95 | | | *p*-value | | | 0.05 | | |
| Information | | | | | | Responsibility | | | | | |
| Female (%) | 5 | 12 | 17 | 32 | 34 | Female (%) | 38 | 30 | 19 | 10 | 3 |
| Male (%) | 5 | 10 | 19 | 41 | 25 | Male (%) | 34 | 28 | 22 | 13 | 2 |
| *p*-value | | | 0.09 | | | *p*-value | | | 0.07 | | |
| Confidence | | | | | | Avoidance | | | | | |
| Female (%) | 3 | 6 | 18 | 46 | 28 | Female (%) | 22 | 21 | 28 | 23 | 6 |
| Male (%) | 1 | 7 | 23 | 44 | 25 | Male (%) | 25 | 25 | 25 | 19 | 5 |
| *p*-value | | | 0.18 | | | *p*-value | | | **0.02** | | |
| Assistance | | | | | | Denial | | | | | |
| Female (%) | 1 | 4 | 11 | 40 | 44 | Female (%) | 32 | 32 | 23 | 11 | 2 |
| Male (%) | 1 | 3 | 17 | 41 | 38 | Male (%) | 27 | 31 | 25 | 14 | 3 |
| *p*-value | | | **0.02** | | | *p*-value | | | **0.03** | | |
| | | | | | | Cost | | | | | |
| | | | | | | Female (%) | 31 | 29 | 22 | 14 | 4 |
| | | | | | | Male (%) | 27 | 29 | 24 | 15 | 5 |
| | | | | | | *p*-value | | | 0.10 | | |

**Table 7.** Two-sided Wilcoxon rank sum test results for the attitudes of males and females towards preparedness. Pros and Cons [0, 1]. Overall attitude [−1, 1].

| | **Male** Mean ± SD | **Female** Mean ± SD | **W** | ***p*-Value** |
|--|--------------------|----------------------|-------|---------------|
| **Pros *"Getting ready is worthwhile"*** | 0.72 ± 0.18 | 0.73 ± 0.19 | 135,463.5 | 0.13 |
| **Cons *"Getting ready is not worthwhile"*** | 0.33 ± 0.23 | 0.31 ± 0.21 | 123,049.5 | 0.23 |
| **Overall attitude (Pros-Cons)** | 0.39 ± 0.32 | 0.42 ± 0.33 | 135,673.5 | 0.12 |

**Risk perception and preparedness**.—A question directly addressed in this study was whether the perceived risk can motivate preparedness. We computed Spearman's rank correlation to assess the relationship between our risk perception results (likelihood, impact, self-efficacy and overall risk perception) for each of the reported hazards and the overall attitudes towards preparedness. We found weak correlations for the gender groups in all cases (rho < 0.20) suggesting that in our study the considered risk factors have a very low association with motivations for preparedness.

**Gender and intersectional factors**.—While gender is important it is shaped by other factors likely to reveal subgroup differences among males and females. We conducted an additional intersectional analysis considering gender related to age and educational background. This analysis revealed interesting findings that emerged during the process of the investigation.

*Gender and age:* We defined six categories for the comparison: YF (young female < 30 years), AF (adult female 30–50 years), OF (Older female > 50 years), YM (young male < 30 years), AM (adult male 30–50 years), OM (older male > 50 years). The mean and standard deviation of risk scores produced by each subgroup are displayed in Table 8. Kruskal-Wallis tests showed statistically significant differences in risk perception between subgroups (Table 8). Pairwise comparisons using Dunn's test indicated that several sub-

groups were observed to be significantly different (Table 9). Interestingly, risk perception for all hazards in females was significantly higher than in males in the same range of age (AF vs. AM and YF vs. YM). We only found significant differences between the same gender in males > 50 years (OM) who perceived higher risks in all hazards than males 30–50 years (AM).

**Table 8.** Mean and standard deviation of risk scores [0, 1] by gender and age and *p*-values from the Kruskal-Wallis test ($\alpha = 0.05$).

| Group | Gender | Age (Years) | Extreme Weather | Fire | Earthquake | Hazard. Mate Accident | Terrorist Attack |
|-------|--------|-------------|-----------------|------|------------|-----------------------|------------------|
| AF | Female | 30–50 | $0.25 \pm 0.19$ | $0.24 \pm 0.16$ | $0.18 \pm 0.16$ | $0.22 \pm 0.18$ | $0.25 \pm 0.21$ |
| OF | Female | >50 | $0.26 \pm 0.18$ | $0.22 \pm 0.15$ | $0.21 \pm 0.17$ | $0.29 \pm 0.22$ | $0.29 \pm 0.24$ |
| YF | Female | <30 | $0.26 \pm 0.19$ | $0.25 \pm 0.17$ | $0.18 \pm 0.16$ | $0.25 \pm 0.19$ | $0.25 \pm 0.20$ |
| AM | Male | 30–50 | $0.20 \pm 0.16$ | $0.18 \pm 0.13$ | $0.13 \pm 0.12$ | $0.20 \pm 0.17$ | $0.21 \pm 0.22$ |
| OM | Male | >50 | $0.23 \pm 0.18$ | $0.22 \pm 0.15$ | $0.19 \pm 0.18$ | $0.24 \pm 0.21$ | $0.25 \pm 0.22$ |
| YM | Male | <30 | $0.19 \pm 0.16$ | $0.18 \pm 0.16$ | $0.13 \pm 0.13$ | $0.20 \pm 0.19$ | $0.20 \pm 0.22$ |
| | *p*-value | | <0.001 | <0.001 | <0.001 | <0.001 | <0.001 |

**Table 9.** Results of pairwise comparison using Dunn's test (z-statistic and *p*-values) for risk perception according to gender and age ($\alpha = 0.05$). Grey cells indicate significant differences between subgroups for all hazards.

| | Extreme Weather | Fire | Earthquake | Hazard. Mate Accident | Terrorist Attack |
|--|-----------------|------|------------|-----------------------|------------------|
| AF vs. OF | −0.52 | 0.71 | −1.57 | −2.13 * | −1.51 |
| AF vs. YF | −0.56 | −0.50 | −0.13 | −1.39 | −0.30 |
| AF vs. AM | 3.43 *** | 4.08 *** | 3.07 ** | 1.90 * | 2.19 * |
| AF vs. OM | 1.46 | 1.35 | −0.40 | −0.12 | −0.18 |
| AF vs. YM | 3.38 *** | 4.58 *** | 3.03 ** | 1.37 | 2.96 ** |
| OF vs. YF | 0.05 | −1.07 | 1.40 | 0.93 | 1.21 |
| OF vs. AM | 3.19 * | 2.51 ** | 3.93 *** | 3.57 *** | 3.19 *** |
| OF vs. OM | 1.65 * | 0.37 | 1.21 | 1.97 * | 1.33 |
| OF vs. YM | 3.24 ** | 3.14 *** | 3.89 *** | 3.02 ** | 3.79 *** |
| YF vs. AM | 3.74 ** | 4.28 *** | 2.98 ** | 3.11 *** | 2.33 * |
| YF vs. OM | 1.90 * | 1.74 * | −0.25 | 1.22 | 0.12 |
| YF vs. YM | 3.68 ** | 4.76 *** | 2.98 ** | 2.47 ** | 3.06 ** |
| AM vs. OM | −1.87 * | −2.60 ** | −3.31 *** | −1.93 * | −2.26 * |
| AM vs. YM | 0.38 | 1.00 | 0.35 | −0.28 | 1.03 |
| OM vs. YM | 2.02 * | 3.28 *** | 3.26 *** | 1.42 | 3.01 ** |

* $p < 0.05$; ** $p < 0.01$; *** $p < 0.001$.

Table 10 displays the mean and standard deviation of preparedness scores and the results of the Kruskal-Wallis test showing significant differences between subgroups (Table 10). Results of the pairwise comparisons by Dunn's test (Table 11) revealed significant intergender differences in the overall measures between adult females (AF) and young males (YM), older females (OF) and adult males (AM), older females (OF) and young males and (YM). We also found differences in the overall measures between the same genders: adult vs. older in females (AF vs. OF), adult vs. older males (AM vs. OM) and older vs. young males (OM vs. YM).

**Table 10.** Mean and standard deviation of preparedness scores by gender and age and *p*-values from the Kruskal-Wallis test ($\alpha = 0.05$). Pros [0, 1] = in favour of preparedness; Cons [0, 1] = against preparedness; Overall [−1, 1] = Overall attitude toward pre-preparedness.

| Group | Gender | Age (Years) | Pros | Cons | Overall |
|---|---|---|---|---|---|
| AF | Female | 30–50 | 0.73 ± 0.19 | 0.33 ± 0.23 | 0.40 ± 0.34 |
| OF | Female | >50 | 0.77 ± 0.19 | 0.27 ± 0.21 | 0.50 ± 0.34 |
| YF | Female | <30 | 0.72 ± 0.18 | 0.31 ± 0.20 | 0.41 ± 0.31 |
| AM | Male | 30–50 | 0.72 ± 0.18 | 0.34 ± 0.23 | 0.38 ± 0.32 |
| OM | Male | >50 | 0.75 ± 0.18 | 0.30 ± 0.22 | 0.45 ± 0.31 |
| YM | Male | <30 | 0.69 ± 0.18 | 0.37 ± 0.22 | 0.32 ± 0.32 |
| | *p*-value | | **<0.01** | **0.012** | **<0.001** |

**Table 11.** Results of pairwise comparison using Dunn's test (z-statistic and *p*-values) for attitudes towards preparedness ($\alpha = 0.05$). Grey cells indicate significant differences between subgroups for all the measures.

| | Pros | Cons | Overall |
|---|---|---|---|
| AF vs. OF | −1.92 * | 1.99 * | −2.52 ** |
| AF vs. YF | 0.00 | 0.76 | −0.39 |
| AF vs. AM | 0.98 | −0.37 | 0.63 |
| AF vs. OM | −1.15 | 1.60 | −1.70 * |
| AF vs. YM | 2.05 * | −1.69 * | 2.33 ** |
| OF vs. YF | 1.83 * | −1.29 | 2.09 * |
| OF vs. AM | 2.64 ** | −2.23 * | 2.95 ** |
| OF vs. OM | 0.95 | −0.67 | 1.11 |
| OF vs. YM | 3.40 *** | −3.16 *** | 4.16 *** |
| YF vs. AM | 0.91 | −1.08 | 0.96 |
| YF vs. OM | −1.08 | 0.76 | −1.21 |
| YF vs. YM | 1.94 * | −2.25 * | 2.54 ** |
| AM vs. OM | −2.04 * | 1.89 * | −2.23 * |
| AM vs. YM | 1.16 | −1.32 | 1.73 * |
| OM vs. YM | 2.95 ** | −2.98 ** | 3.68 *** |

* *p* < 0.05; ** *p* < 0.01; *** *p* < 0.001.

*Gender and Education*: We categorized the sample into six subgroups according to gender and education background: PF (primary female), SF (secondary female), UF (university female), PM (primary male), SM (secondary male), UM (university male). There were significant differences between the subgroups (Tables 12 and 13). We found that risk perception for all hazards in females was significantly higher than in males at the same education level, i.e., secondary (SF vs. SM) and university (UF vs. UM). We also found that females with secondary education (SF) perceived risk to be significantly higher than their gender counterparts with university education (UM).

The Kruskal-Wallis test showed significant differences only between subgroups by gender and education for the overall attitudes toward preparedness (Table 14). Pairwise comparisons showed significant intergender differences for UM vs. PF (*p* = 0.012), UF vs. PM (*p* = 0.018) and UF vs. SM (*p* < 0.01). We also found differences between subgroups of the same gender: SF vs. PF (*p* = 0.012), and UF vs. PF (*p* < 0.01).

**Table 12.** Mean and standard deviation of risk scores [0, 1] by gender and education and *p*-values from the Kruskal-Wallis test ($\alpha$ = 0.05).

| Group | Gender | Education | Extreme Weather | Fire | Earthquake | Hazard. Mate Accident | Terrorist Attack |
|---|---|---|---|---|---|---|---|
| PF | Female | Primary | 0.22 ± 0.21 | 0.25 ± 0.16 | 0.20 ± 0.16 | 0.24 ± 0.17 | 0.23 ± 0.19 |
| FS | Female | Secondary | 0.25 ± 0.18 | 0.22 ± 0.15 | 0.18 ± 0.17 | 0.23 ± 0.17 | 0.24 ± 0.20 |
| FU | Female | University | 0.26 ± 0.18 | 0.25 ± 0.17 | 0.17 ± 0.15 | 0.25 ± 0.20 | 0.26 ± 0.22 |
| PM | Male | Primary | 0.18 ± 0.14 | 0.16 ± 0.09 | 0.15 ± 0.12 | 0.19 ± 0.17 | 0.20 ± 0.20 |
| SM | Male | Secondary | 0.19 ± 0.15 | 0.19 ± 0.14 | 0.15 ± 0.15 | 0.21 ± 0.19 | 0.21 ± 0.22 |
| UM | Male | University | 0.22 ± 0.18 | 0.20 ± 0.16 | 0.15 ± 0.15 | 0.22 ± 0.19 | 0.23 ± 0.22 |
| | *p*-value | | <0.001 | <0.001 | 0.013 | 0.034 | 0.038 |

**Table 13.** Results of pairwise comparison using Dunn's test (z-statistic and *p*-values) for risk perception according to gender and education background ($\alpha$ = 0.05).

| | Extreme Weather | Fire | Earthquake | Hazard. Mate Accident | Terrorist Attack |
|---|---|---|---|---|---|
| PF vs. SF | −1.97 * | 0.93 | 0.64 | 0.27 | −0.22 |
| PF vs. UF | −2.08 * | 0.34 | 0.69 | −0.00 | 0.34 |
| PF vs. PM | 0.33 | 2.81 ** | 0.81 | 1.47 | 1.02 |
| PF vs. SM | 0.40 | 2.48 ** | 1.93 * | 1.48 | 1.15 |
| PF vs. UM | −0.71 | 2.19 * | 2.05 * | 0.96 | 0.57 |
| SF vs. UF | −0.14 | −1.10 ** | 0.06 | −0.52 | −0.22 |
| SF vs. PM | 2.60 ** | 2.76 ** | 0.39 | 1.68 * | 1.61 |
| SF vs. SM | 4.27 *** | 2.76 ** | 2.30 * | 2.16 * | 2.45 ** |
| SF vs. UM | 2.33 ** | 2.27 * | 2.55 ** | 1.25 | 1.45 |
| UF vs. PM | 2.73 ** | 3.47 *** | 0.35 | 2.02 * | 1.78 * |
| UF vs. SM | 4.66 *** | 4.05 *** | 2.36 ** | 2.82 ** | 2.83 ** |
| UF vs. UM | 2.62 ** | 3.56 *** | 2.63 ** | 1.87 * | 1.77 * |
| PM vs. SM | −0.02 | −1.10 | 1.00 | −0.37 | −0.13 |
| PM vs. UM | −1.23 | −1.44 | 1.13 | −0.95 | −0.76 |
| SM vs. UM | −2.06 * | −0.56 | 0.20 | −0.97 | −1.07 |

* $p < 0.05$; ** $p < 0.01$; *** $p < 0.001$.

**Table 14.** Mean and standard deviation of preparedness scores by gender and education and *p*-values from the Kruskal-Wallis test ($\alpha$ = 0.05). Pros [0, 1] = in favor of preparedness; Cons [0, 1] = against preparedness; Overall [−1, 1] = Overall attitude toward preparedness. The significant *p*-value is in bold.

| Group | Gender | Education | Pros | Cons | Overall |
|---|---|---|---|---|---|
| PF | Female | Primary | 0.64 ± 0.26 | 0.37 ± 0.23 | 0.27 ± 0.41 |
| FS | Female | Secondary | 0.73 ± 0.18 | 0.32 ± 0.22 | 0.41 ± 0.32 |
| FU | Female | University | 0.74 ± 0.17 | 0.29 ± 0.20 | 0.44 ± 0.31 |
| PM | Male | Primary | 0.70 ± 0.19 | 0.35 ± 0.21 | 0.35 ± 0.31 |
| SM | Male | Secondary | 0.72 ± 0.19 | 0.34 ±0.23 | 0.37 ± 0.33 |
| UM | Male | University | 0.72 ± 0.16 | 0.31 ± 0.21 | 0.41 ± 0.30 |
| | *p*-value | | 0.106 | 0.083 | **0.019** |

## 4. Discussion

This exploratory study looks at gender differences in two central aspects (1) risk perception of multiple hazards and (2) attitudes towards disaster preparedness. People (*n* = 1.014) from different countries (Italy, Sweden, Poland, France and Spain) were included in this study.

**Risk perception**.—Our results indicate gender differences in concerns about hazards expressed as the likelihood of the occurrence, personal consequences and self-efficacy. Females were more aware of the occurrence of disasters resulting from extreme weather and fire. Females also exhibited a higher perception of the potential consequences of extreme weather, fire and earthquake than males. These results are in line with previous findings confirming that women are more worried about natural hazards than men, especially if family members are threatened [10,42]. It is important to note that items did not necessarily have the same meaning for females and males as they may give priority to different hazards and/or show different concerns about the same hazards [43]. The questions of this section included "what in your view is the impact for you and your family?" Female respondents perhaps felt more oriented towards home and family when they thought about the presented hazards. Following this type of interpretation would yield an explanation as to why female respondents showed higher concerns. Our results reinforce a recent finding that reports no gender effects in the perceived vulnerability regarding terrorist attacks [44].

Self-efficacy has been identified as an important variable that should be considered within the context of hazards research, since it may be linked to the perceived risk and the adoption of hazard adjustments [45]. Our study confirms that gender is an important factor in the perceived self-coping abilities to deal with disasters. Males reported higher self-efficacies for all the presented hazards (i.e., extreme weather, fire, earthquake, hazardous accident and terrorist attack). A possible explanation suggests that women may be less confident than men, but this conceivably denotes a more realistic view of their own self-capacities [13].

To measure the overall risk perception (R) for the five hazards we computed the three subjective factors, namely: the likelihood of a disaster to occur (L), the impact if a disaster occurs nearby (I) and the perceived self-efficacy to cope with the disaster (E) through the following equation R = (L × I)/E. The resulting scores of males and females were compared. Our results showed that the risks were judged significantly higher by females in all hazards. The main reason that explains these results may be associated with the incorporation of self-efficacy (E) by assuming that this factor affects the risk perceived rather than simply considering the perceived likelihood (L) and impact (I) to measure risk ratings [17]. Additionally, we computed this approach (i.e., R = L × I) and the differences were also statistically significant (females scored higher risk than males) when applying two-sided Wilcoxon rank sum test for extreme weather conditions (W = 140,553.5, *p* < 0.001), fire (W = 141,096, *p* < 0.001) and earthquake (W = 138,242, *p* = 0.03). But the differences were not significant for hazardous materials accidents (W = 131,362, *p* = 0.53) and terrorist attacks (W = 129,487, *p* = 0.83). Overall, this finding emphasises gender differences in risk perception for natural hazards. But this might not be the case for man-made hazards, thus warranting further research to explore the factors that may account for it.

**Attitudes towards preparedness.**—Participants were also asked about the Pros and Cons of preparedness to measure their individual interest in getting prepared. Responses to some items differed between males and females. The importance of being prepared to help others (Assistance) was significantly higher in female respondents. This result was in line with previous studies attesting that women tend to be more altruistic than men (see for some references [46–49]. The statement that disasters "don't happen where I live" (Denial) had significantly higher scores in male respondents denoting differences in judgments based upon such events [43] and optimistic bias [50]. The 'It will not happen to me' belief is a very important aspect of preparedness that has already been reported [51] since overconfidence can keep individuals from realizing how little they know and how much information they may need to be ready. By contrast, females were significantly

more likely to "not think about bad things happening" (Avoidance) than males. This result supports previous studies attesting that gender is a significant predictor of coping through avoidance [52,53]. Avoidance here can be associated with information avoidance leading to misinformation which has been recently analysed in the context of the COVID 19 pandemic [54,55]. More research is needed to explore gender influence on this aspect of behaviour in the context of disasters.

**Risk perception and preparedness.**—A question addressed in this study was whether the perceived risk can motivate preparedness. Risk perception has been considered a predictor or correlates of preparedness behaviour. While some studies found that risk perception predicted or was associated with preparedness others found no effects [27]. Overall, results showed that most respondents (both males and females) were in favour of preparedness. More importantly, despite the gender differences in risk perception, there were no significant differences in attitudes towards preparedness. We found weak correlations for the gender groups in all cases (rho < 0.20) suggesting that in our study the considered risk factors have a very low association with motivations to seek preparedness.

**Gender and intersectional factors.**—Previous results open a new question addressed in this study. There are differences/similarities between women and men but which women? and which men? An intersectional approach was used to analyze the multiplicative impact of gender when combined with age and educational background which also shapes the identity, perceptions, and attitudes of individuals towards disasters. This analysis produced more detailed information showing the differences between different but interdependent categories and factors. We compared six subgroups categorized by gender (male; female) and age (<30 years; 30–50 years; >50 years). The subgroups were observed to be significantly different, especially between subgroups of different genders. Young and adult females perceived higher risks than their gender counterparts of the same age. There were also gender differences in preparedness between subgroups of different ages denoting that age is also a contributing factor to people's attitudes (e.g., females in a higher age range are more motivated for preparedness than men in a lower age range). We defined six subgroups according to gender (male; female) and education (primary; secondary; university). We found that risk perception for all hazards in females was significantly higher than in males at the same education level (e.g., secondary and university education background). Furthermore, females with secondary education (SF) perceived risks to be significantly higher than their gender counterparts with university education (UM). When it comes to the attitudes towards preparedness, we found no significant differences between subgroups in the reported scores for the pros and cons of getting ready for disasters. However, females at a higher level of education have more positive attitudes towards preparedness.

The current study has several strengths. First, it contributes to the literature by providing a general approach to exploring gender on public perception of multiple hazards/disasters, which predominantly has been concerned with specific disasters and affected communities. Second, the datasets generated in this study are available allowing third parties to conduct further research. Third, this study proposes a new approach to computing the subjective factors (likelihood, impact and self-efficacy) of risk perception and personal attitudes toward preparedness (Pros and Cons). Fourth, the resulting scores were presented in more analytical forms, i.e., [0, 1] and [−1, 1] for better understanding and further comparison with other datasets. Fifth, the intersectional analysis presented here (combining gender age and education) will allow policy-makers and first responders to better identify subgroups and factors among the population to implement DRR policies and practices. Finally, as mentioned the results produced here not only have scientific value but also have the potential to inform EU policymakers and first responders. The current study also has its limitations. First, compared with other studies a "small" sample size was employed (*n* = 1.014). However, we believe that the subset of the population used was representative thus providing a sufficient amount of information to conclude the EU population. Second, the study is limited in scope, i.e., whether or not gender is "significant" along with other two factors (i.e., age and education level). Third, the association between

risk perception and preparedness was not found, perhaps due to the design of the items in the questionnaire as this was not the initial purpose of the study. Fourth, non-parametric methods were conducted for the statistical analysis potentially leading to less powered results. Finally, the results presented here are rather indicative than definitive.

## 5. Conclusions

Results presented in this study constitute the first process of gender analysis (e.g., data collection, data processing, and analysis) and advocate to conduct of the second process which is interpretative in nature [56]. The gender discrepancies may reveal the underlying mechanisms apart from biological and physiological differences such as everyday life behaviours and beliefs as well as stereotypes derived from gender norms [57]. Conceivably, socioeconomic and cultural differences between men and women are more evident in lower-income countries leading to a higher exposure of women to risks in case of a disaster [5]. The present results suggest that gender differences in relation to risk perception of multiple hazards might still be present in European societies. The different social roles and activities of men and women within the household and community are examples of how gender norms and ideals manifest. The role of nurturer and caregiver primarily played by women has been associated with greater concern about the risk of potential disasters and the well-being of others [58]. Also, different gender roles can be reinforced in disasters because expectations for men and women are usually based on stereotypes. For instance, a recent study focused on actions during a large Swedish forest fire, indicated that women were praised when they followed the traditional norms but denigrated when they performed what was perceived as male-coded tasks [59].

Our results suggest the same predisposition of females and males to seek preparedness. Women are slightly present in emergency planning and disaster management programs but more involved in household and community care in practice [57,60,61] and often ignored in official evaluations after disasters [5]. It is argued here that gender skills may benefit the prevention and mitigation of hazard situations.

Although limited to risk perception and preparedness, the outcomes of this study can provide insights into the integration of gender-sensitive practices in disaster preparedness and response. First, conducting more qualitative and quantitative research to better understand gender-based roles and responsibilities is highly desirable. For studying a complex area such as gender constructs and roles, multi-disciplinary research could be beneficial. Second, improving women's capacities and knowledge (training and education) can increase individual and community resilience. Third, promoting policies and actions to involve women in official emergency management programs and decision-making is essential to minimize gender gaps in disaster planning and response. Much work remains to be done to systematically integrate gender analysis into relevant domains of safety science and technology—from strategic considerations for establishing research priorities to guidelines for establishing best practices in formulating research questions, designing methodologies and interpreting data.

The practical implications of this study can be summarized as follows:

- Datasets produced here are available to practitioners, policymakers and first responders to conduct further analyses on gender in DRR across the EU population.
- The present study has gone one step beyond gender analysis by providing information on risk perception and motivation for disaster preparedness across different population subgroups (individual-level categories while considering gender). Therefore, the interested parties will be able to focus on promoting specific DRR policies and practices (bottom-up participatory and learning processes) through interventions oriented to specific target groups.

**Author Contributions:** Conceptualization, A.C. (Arturo Cuesta), D.A. and A.C. (Antonio Carnevale); data curation, A.C. (Arturo Cuesta); Formal analysis, A.C. (Arturo Cuesta); Funding acquisition, all authors; investigation, A.C. (Arturo Cuesta); methodology, A.C. (Arturo Cuesta), D.A. and A.C. (Antonio Carnevale); Project administration, D.A.; supervision, A.C. (Arturo Cuesta) and D.A.; writing—original draft, A.C. (Arturo Cuesta); writing—review & editing, F.A. All authors have read and agreed to the published version of the manuscript.

**Funding:** This project has received funding from the European Union's Horizon 2020 research and innovation programme under grant agreement No 832576.

**Institutional Review Board Statement:** The study was conducted in accordance with the Declaration of Helsinki and approved by the Ethics Committee of the University of Cantabria (CE Proyecto 06/2019).

**Informed Consent Statement:** Informed consent was obtained from all subjects involved in the study.

**Data Availability Statement:** The datasets in this study are available from the corresponding author upon request.

**Acknowledgments:** The authors would like to thank the European Commission and the members of the ASSISTANCE consortium for their collaboration and support to this study. We would like to express our very great appreciation to the team members involved in the translation of the questionnaire into Italian (CYBERETHICS LAB SRLS), Swedish (RISE RESEARCH INSTITUTES OF SWEDEN AB), Polish (PRZEMYSLOWY INSTYTUT AUTOMATYKI I POMIAROW PIAP), French (THALES SA) and Spanish (UNIVERSIDAD DE CANTABRIA).

**Conflicts of Interest:** The authors declare no conflict of interest. The funders had no role in the design of the study; in the collection, analyses, or interpretation of data; in the writing of the manuscript, or in the decision to publish the results.

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
