# Peer review of "Gender and Public Perception of Disasters: A Multiple Hazards Exploratory Study of EU Citizens"

_safety, 2022_

Round 1

Reviewer 1 Report

This paper reports on the methods for gender analysis, briefly summarizes the key findings and draws conclusions about the potential impact of gender on risk perception and preparedness for disasters. Generally, this paper is scientifically sound and novel. However, I cannot recommend the work for publication in the current form unless the following concerns are clearly and accurately addressed:

1.       There is no doubt that the different disaster contributes to a different level of risk perception. Therefore, the types and consequences of specific disasters should be presented instead of showing an overall statistic in the introduction. Moreover, the reason why the authors choose these disasters and the consistency of the impact of disasters on different genders should be explained.

2.       The authors should elaborate more on the basis of the questionnaire survey in the five member states of the EU, and whether the data volume in the article is sufficient for representing the overall level of the EU citizens.

3.       The equation R= (L x I)/E is the foundation of the whole work, please elaborate more on it.

4.       The reviewer thinks that the job of the participant matters. A firefighter or a soldier will be less afraid of disasters compared to an office staff. At least whether the job is related to accidents or emergency or extreme environments should be considered and stated in the manuscript.

5.       There is a large age gap between male and female participants in this paper, and further analysis of gender differences in different age ranges can be considered in the results and discussions.

6.       What are the reasons for using different assessment scales for likelihood, impact and self-efficacy.

7.       The reference format needs to be checked and unified.

Reviewer 2 Report

This paper analyzes the differences between men and women in risk rating for each specific hazard and preparedness. It is necessary to consider the following points.

First, the difference in risk perception between men and women has already been verified in many studies. It is necessary to clearly present what differentiates this study.

Second, the two topics of risk perception and preparedness are being analyzed simultaneously. It seems to be two papers. It is necessary to strengthen the theoretical discussion so that it can be viewed as a single thesis by strengthening the link between the two themes.

Third, it is necessary to add to the conclusion part what practical implications this study can give.

Fourth, can't we show differences within women? There will be differences in the  variables depending on age, educational background, and income among women.

Reviewer 3 Report

Thank you to the authors for this interesting and well written manuscript. I think it adds tremendous value to our understanding of gender differences in both risk perception as well as disaster preparedness. This is a topic that is very familiar to me and the authors have made an important contribution here by filling in gaps on gender differences. 

The format, writing, background, methods, results ,discussion are all very high quality. I only had a bit of a problem reading and following Figure 1. If there is no simpler way to show these data then I would leave it alone.  

The sample is impressive. Conclusions are justified and clear.

I congratulate the authors on this high quality work and manuscript.

Reviewer 4 Report

Overall, the paper is clearly written and can be published after some necessary changes are incorporated into the revised manuscript. The suggestion to the author(s) is directly inserted in the pdf version.

Reviewer 5 Report

Dear Authors,

The review of the study titled "Gender and public perception of disasters: A multiple hazards exploratory study of EU citizens" has been completed. The article is well structured and well written. I think that the study will contribute to the literature.

In order to make the work more beautiful and readable, the following minor corrections should be made:

- in line 122, the word "responses" is incorrectly split at the end of the sentence.

- in line 388, the (-) in the word "sci-ence" should be deleted

- in line 401, the (-) in the word "per-ception" should be deleted

- in line 402, the (-) in the word "International" should be deleted

- in line 404, the (-) in the word "Prepared-ness" should be deleted

- in line 418, the (-) in the word "Haz-ards" should be deleted

In addition, the following resources can be used to contribute to the study:

- Cvetković, V. M., Öcal, A., & Ivanov, A. (2019). Young adults’ fear of disasters: A case study of residents from Turkey, Serbia and Macedonia. International Journal of Disaster Risk Reduction35, 101095.

Round 2

Reviewer 1 Report

The current vesion can be accepted.

Reviewer 2 Report

Authors revised all of things reviewer commented

Reviewer 4 Report

The author(s) has addressed all the suggested changes. Therefore i accept the paper in the current form

This manuscript is a resubmission of an earlier submission. The following is a list of the peer review reports and author responses from that submission.